# Verification of Photometric Parallaxes with Gaia DR2 Data

Oleg Malkov [1,*], Sergey Karpov [2,3,4], Dana Kovaleva [1], Sergey Sichevsky [1],
Dmitry Chulkov [1], Olga Dluzhnevskaya [1], Alexey Kniazev [3,5,6,7], Areg Mickaelian [8],
Alexey Mironov [7], Jayant Murthy [9], Alexey Sytov [1], Gang Zhao [10]
and Aleksandr Zhukov [1,7,11]

[1] Institute of Astronomy of RAS, 119017 Moscow, Russia; dkovaleva@mail.ru (D.K.);
   s.sichevskij@gmail.com (S.S.); chulkov@inasan.ru (D.C.); olgad@inasan.ru (O.D.); sytov@inasan.ru (A.S.);
   aozhukov@mail.ru (A.Z.)
[2] Institute of Physics, Czech Academy of Sciences, 182 21 Prague, Czech Republic; karpov.sv@gmail.com
[3] Special Astrophysical Observatory of RAS, 36916 Nizhnij Arkhyz, Russia; akniazev@saao.ac.za
[4] Laboratory "Fast Variable Processes in the Universe", Kazan Federal University, 420008 Kazan, Russia
[5] South African Astronomical Observatory, P.O. Box 9, Cape Town 7935, South Africa
[6] Southern African Large Telescope Foundation, P.O. Box 9, Cape Town 7935, South Africa
[7] Sternberg Astronomical Institute, 119234 Moscow, Russia; almir@sai.msu.ru
[8] NAS RA V. Ambartsumian Byurakan Astrophysical Observatory, Byurakan 0213, Republic of Armenia;
   aregmick@yahoo.com
[9] Indian Institute of Astrophysics, Bengaluru 560 034, India; jmurthy@yahoo.com
[10] Key Laboratory of Optical Astronomy, National Astronomical Observatories, Chinese Academy of Sciences,
   Beijing 100012, China; gzhao@nao.cas.cn
[11] Russian Technological University (MIREA), 119454 Moscow, Russia
* Correspondence: malkov@inasan.ru; Tel.: +7-495-951-7993

**Abstract:** Results of comparison of *Gaia* DR2 parallaxes with data derived from a combined analysis of 2MASS (Two Micron All-Sky Survey), SDSS (Sloan Digital Sky Survey), GALEX (Galaxy Evolution Explorer), and UKIDSS (UKIRT Infrared Deep Sky Survey) surveys in four selected high-latitude $|b| > 48°$ sky areas are presented. It is shown that multicolor photometric data from large modern surveys can be used for parameterization of stars closer than 4400 pc and brighter than $g_{SDSS} = 19.^{m}6$, including estimation of parallax and interstellar extinction value. However, the stellar luminosity class should be properly determined.

**Keywords:** parallax; photometry; interstellar extinction; surveys; Gaia

## 1. Introduction

One of the main problems of astrophysics is the study of the physical properties belonging to the surface layers of stars. Because these stars are observed through interstellar dust, their light is dimmed and reddened, complicating their parameterization and classification. The parameters of a given star (temperature, gravity, metallicity, etc.), as well as the interstellar reddening, may be obtained from its optical, infrared, and ultraviolet spectrum. However, one must either use large telescope or bright objects to get spectral energy distributions with good resolution and sufficient accuracy. For instance, spectroscopic observations with 1-h exposure time on a 2-m telescope with low ($R \sim 1000$) and high ($R \sim 100,000$) resolution allow limiting magnitudes of 16–17 mag and 11–12 mag, respectively. An 8-m telescope adds 3 mag to these estimations. This work was performed by various authors, and a number of empirical atlases were constructed (Straizys and Sviderskiene [1], Glushneva et al. [2],

Alekseeva et al. [3], Alekseeva et al. [4], Pickles [5], Bagnulo et al. [6], Le Borgne et al. [7], Valdes et al. [8], Heap and Lindler [9], Falcón-Barroso et al. [10], Wu et al. [11]). However Mironov et al. [12] made a critical analysis, compared data for stars included in several atlases, and found many discrepancies.

Another way to construct a map of interstellar extinction is its estimation (as well as stellar parameters) from evolutionary tracks. Corresponding procedures were developed in Sichevsky and Malkov [13], Sichevskij [14], Sichevskij [15] and applied to LAMOST data by Sichevskij [16]. However, a knowledge of stellar atmospheric parameters is highly desirable for the application of these procedures, limiting the number of stars available for such a parameterization.

Therefore, the solution of the problem of parameterization of stars based on their photometry is a topical issue [17]. A great variety of photometric systems (see, e.g., Straižys [18] for references) and recently constructed large photometric surveys (like SDSS [19] and GALEX [20]) as well as VO-tools for cross-matching surveys' objects provide with a possibility to get multicolor photometric data for millions of objects. Consequently, it allows the user to parameterize objects and determine interstellar extinction in the galaxy.

A comparative analysis of available 3D maps of interstellar extinction was made by Kilpio and Malkov [21], and contradictory results were found. As a temporary solution of that problem, a synthetic map of the galactic interstellar extinction can be compiled (see, e.g., [22–24]).

Early dust maps used the correlation between the dust column density and the distribution of neutral hydrogen [25]. These data were supplanted by the dust maps produced by Schlegel et al. [26], who used full sky microwave data made available by the IRAS (Infrared Astronomical Satellite) mission and the DIRBE (Diffuse Infrared Background Experiment) instrument on the COBE (Cosmic Background Explorer) mission. Mapping the dust column densities via the calibrated dust temperature, the extinction maps, assuming a standard reddening law, were shown to be at least twice as accurate as those of Burstein and Heiles [25]. An advantage of this method is that it does not rely on a predefined model for the stellar population.

The successful implementation of the European astrometric space mission *Gaia* (the second version of the mission catalog, *Gaia* DR2, was published in April 2018 [27]) allows the solving of several stellar astronomy problems, like determination of stellar mass, age estimation and others. In particular, it became possible to improve the results of the parameterization of stars, carried out from multicolor photometry.

In this paper, the verification of the method and stellar sample analyzed in Malkov et al. [28] using *Gaia* DR2 data is described. We also discuss how including the *Gaia* parallaxes into the procedure would improve the accuracy of parameterization, and how to select/process objects with unknown *Gaia* parallax for parameterization.

This paper is organized as follows. Data and methods are described in Section 2, Section 3 contains results and conclusions.

## 2. Data and Method

In Malkov et al. [28] (hereinafter—Paper18) objects in four selected areas in the sky were cross-matched (see details of the procedure in [29–31]) with 2MASS (Two Micron All-Sky Survey) [32], SDSS (Sloan Digital Sky Survey) [19], GALEX (Galaxy Evolution Explorer) [20], and UKIDSS (UKIRT Infrared Deep Sky Survey) [33] surveys, and multi-wavelength photometric data were used to determine the parameters of stars. The galactic coordinates of the areas are (334, +61.9), (257, +48.7), (301, +62.1), and (129, −58.1). For the studied objects MK (Morgan-Keenan) spectral types (SpT), distances ($d$) and interstellar extinction values ($A_V$) were estimated, minimizing the function

$$\chi^2 = \sum_{i=1}^{N} \left( \frac{m_{obs,i} - m_{calc,i}}{\sigma m_{obs,i}} \right)^2, \tag{1}$$

where $m_{obs,i}$ and $\sigma m_{obs,i}$ are the apparent magnitude and its observational error, respectively, in the $i$-th photometric band from a given survey, and the summation is over up to $N = 14$ photometric bands (JHK$_S$ from 2MASS, ugriz from SDSS, FUV and NUV from GALEX, YJHK from UKIDSS), and

$$m_{calc,i} = M_i + 5\log d - 5 + A_i, \tag{2}$$

or

$$m_{calc,i} = M_i - 5\log \varpi - 5 + A_i. \tag{3}$$

Here $A_i = f(A_V)$ is the extinction in the i-th photometric band, and can be determined from the interstellar extinction law. To retrieve $A_i$ from $A_V$ we have used data from Yuan et al. [34] for 2MASS [32], SDSS [19] and GALEX [20], whereas data for UKIDSS [33] and Johnson V-band were adopted from Schlafly and Finkbeiner [35]. Both teams made $A_i$ calculations for SDSS photometry, and our comparison shows a very good agreement between their results (see Paper18 for details).

$M_i = f(\text{SpT})$ is the absolute magnitude in i-th photometric band taken from calibration tables. To obtain absolute magnitudes for stars of different spectral types in the corresponding photometric systems $M_i$, we have compiled a table of absolute magnitudes in 2MASS, SDSS and GALEX surveys using Kraus and Hillenbrand [36], and Findeisen et al. [37] data. Due to lack of published data, UKIDSS absolute magnitudes were calculated from 2MASS magnitudes and 2MASS-UKIRT relations from Hodgkin et al. [38].

The distance $d$ and parallax $\varpi$ are expressed in parsec and arcsec, respectively. It should be noted that Equation (3) is not correct when using observational data. The mean value of the parallax is not enough and their errors should be considered to derive a good value for the distance to be substituted in Equation (2) to derive Equation (3) (see Bailer-Jones et al. [39]).

Altogether 251 objects were found in at least three of the four surveys and cross-matched in the four areas, but only 26 of them were successfully parameterized. The following reasons to remove objects from further consideration were considered.

First, the original surveys contain various flags which allow us to remove unsuitable objects, namely, "Binary object" (2MASS, UKIDSS), "Non-stellar/extended object" (2MASS, SDSS, GALEX, UKIDSS), "Observation of low quality" (SDSS). Secondly, overly bright objects and objects with large observational errors were not considered.

Also, only areas located at relatively high galactic latitudes ($|b| > 48°$) are considered in this work. Consequently, $A_V$ is assumed to be smaller than 0.5 mag, and distance $d$ is assumed to be closer than 8000 pc. Objects which presented larger values for those parameters were removed from further consideration.

Finally, for every object a rough parameterization was performed (based on 2MASS+SDSS photometry only) with Covey et al. [40] absolute magnitude tables. An object was removed if this procedure showed that there was a high probability for it to be a non-MS star (giant or supergiant). It should be reminded that in the current study only MS (Main-Sequence) stars are considered.

A comparison of our results for 26 selected stars with independent results obtained from the LAMOST (Large Sky Area Multi-Object Fiber Spectroscopic Telescope) [41] for some of the studied stars has demonstrated a good agreement. Interstellar extinction as a function of distance ($A_V(d)$) was constructed for the four selected areas (see Figures 1–4 in Paper18). These relations were extrapolated to infinity ($d \to \infty$). Please note that the extrapolation is formal, and can introduce some uncertainty. For the resulting $A_V(\infty)$ values in three of the four areas (Nos 1, 5, and 6 in Paper18), a good agreement was found with the data used in the study of supernovae [42] to confirm the accelerated expansion of the Universe. For the remaining area, No 2, an agreement was not achieved (see Table 2 in Paper18).

To identify the conditions under which stars are properly parameterized from multicolor photometry, we relied upon the results obtained in our pilot study of interstellar extinction in four areas (see Paper18), and *Gaia* data. A cross-matching of Paper18 objects with *Gaia* DR2 catalog was made. Among 251 objects, studied in Paper18, only 72 were found in *Gaia* DR2 (such a very small

fraction is understandable, as about 80% of the Paper18 objects are fainter than $19.^m5$ $g_{SDSS}$), and seven of them have no *Gaia* parallaxes. Among 26 objects, selected in Paper18 for construction of $A_V(d)$ relation, one is absent in *Gaia* DR2, and two more objects have no parallaxes.

　　　The *Gaia* trigonometric parallaxes (hereafter $\varpi_{tr}$) are compared with photometric parallaxes obtained in Paper18 (hereafter $\varpi_{ph}$). When analyzing *Gaia* data, the recommendations published in [43] were considered. In particular, the filters, designed on the basis of the photometric and astrometric flags contained in *Gaia* DR2, were taken into account. They were used to construct a so-called astrometrically clean sample, hereafter ACS.

　　　Photometric passbands used in SDSS [44] and *Gaia* DR2 [45] are presented in Figure 1. In this work, we deal with $g_{SDSS}$ and *Gaia* G, BP, and RP magnitudes of the studied stars.

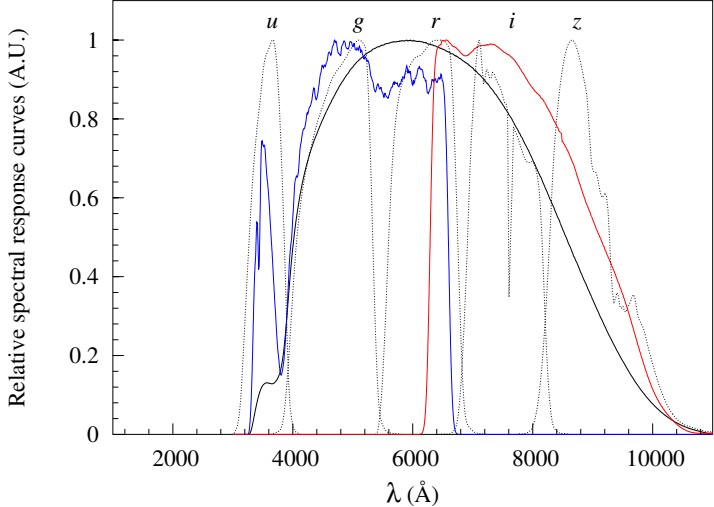

**Figure 1.** SDSS ugriz (gray dashed curves) and *Gaia* G (black solid curve), $G_{BP}$ (blue solid curve), and $G_{RP}$ (red solid curve) photometric passbands.

## 3. Results and Conclusions

　　　A comparison of the photometric and trigonometric parallaxes of stars used in Paper18 for the $A_V(d)$ construction is shown in Figure 2. The ratio between the difference in parallax $\varpi_{ph} - \varpi_{tr}$ and the photometric parallax $\varpi_{ph}$ as a function of $\varpi_{ph}$ and $g_{SDSS}$ is shown in Figure 3. In the current study observational photometric errors are considered to be the only source for the resulting parameters errors. Consequently, here we underestimate the error values. To calculate errors more correctly, one should also consider calibration tables errors and relations errors.

　　　Most stars demonstrate a satisfactory agreement but with some outliers which require an explanation. The Hertzsprung-Russell diagram (HRD) was constructed for *Gaia* DR2 stars with trigonometric parallax $\varpi_{tr} > 10$ mas, with relative parallax uncertainty $\sigma\varpi_{tr}/\varpi_{tr} < 10\%$, with relative error of BP and RP fluxes better than 10%, and satisfying the ACS requirements (Figure 4). The studied stars were added to the plot, and their positions on HRD were analyzed. Data for the studied stars are presented in Table 1.

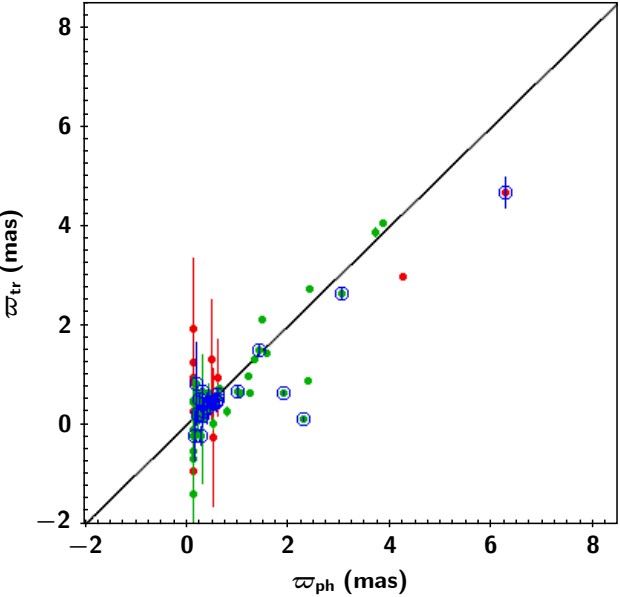

**Figure 2.** Photometric parallax ($\varpi_{ph}$) from Paper18 and trigonometric parallax ($\varpi_{tr}$) from *Gaia* for all stars in common to Paper18 and *Gaia* DR2 that satisfy (green points) and do not satisfy (red points) the ACS requirements. Blue circles are the stars used in Paper18 for the $A_V(d)$ construction. "$\varpi_{tr} = \varpi_{ph}$" is indicated as a black line.

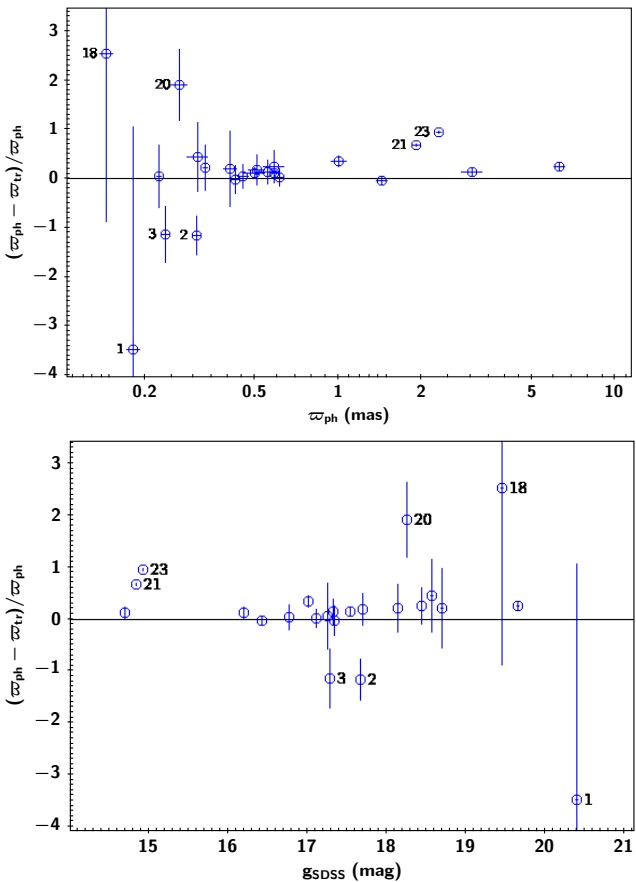

**Figure 3.** The ratio between the difference in parallax $\varpi_{ph} - \varpi_{tr}$ and the photometric parallax $\varpi_{ph}$ vs. $\varpi_{ph}$ (**upper** panel) and vs. $g_{SDSS}$ (**lower** panel) for the stars used in Paper18 for the $A_V(d)$ construction. Labels correspond to running numbers of stars in Table 1.

**Table 1.** Stars used in Paper18 for the $A_V(d)$ construction. Upper part: running number, SDSS and *Gaia* identifiers, RA, and DEC (2000) coordinates. Lower part: running number, SDSS ($g$) and *Gaia* ($G$) photometry with errors, photometric and trigonometric parallax with errors, spectral type estimated in Paper18 and corresponding $M_{bol}$.

| N | SDSS | *Gaia* DR2 | RA2000 (deg) | DEC2000 (deg) |
|---|---|---|---|---|
| 1 | 752-301-2-0325-0286 | 3683644191076349312 | 192.1701 | −0.8165 |
| 2 | 6005-301-2-0158-0092 | 3689648868888993920 | 192.2168 | −0.7911 |
| 3 | 6005-301-2-0158-0061 | 3836643881838700800 | 192.1728 | −0.8335 |
| 4 | 1241-301-1-0173-0042 | 3801228953847961600 | 164.2724 | −3.5562 |
| 5 | 752-301-2-0324-0052 | 3683656594941905920 | 192.0763 | −0.7852 |
| 6 | 1462-301-4-0543-0082 | 3665039324758041600 | 206.7882 | 2.4288 |
| 7 | 6121-301-2-0157-0087 | 3683657110337986048 | 192.1002 | −0.7568 |
| 8 | 6121-301-2-0157-0037 | 3683644569033472512 | 192.1050 | −0.8108 |
| 9 | 1462-301-4-0543-0066 | 3665227032008658560 | 206.7471 | 2.4403 |
| 10 | 2194-301-1-0361-0007 | 3801228683265104640 | 164.1956 | −3.5772 |
| 11 | 1241-301-1-0173-0102 | 3801228610250558848 | 164.2084 | −3.5842 |
| 12 | 7727-301-3-0174-0108 | 2552008097711479808 | 16.2328 | 4.5274 |
| 13 | 2194-301-1-0361-0119 | 3801227819976566272 | 164.2968 | −3.5978 |
| 14 | 752-301-2-0325-0136 | 3689650277638267392 | 192.1738 | −0.7430 |
| 15 | 1462-301-4-0543-0182 | 3665226581036533504 | 206.7508 | 2.4200 |
| 16 | 756-301-1-0510-0121 | 3683643679975511808 | 192.1394 | −0.8415 |
| 17 | 7727-301-3-0174-0122 | 2552021880262020608 | 16.2512 | 4.6030 |
| 18 | 1462-301-4-0544-0120 | 3665040626132812800 | 206.8532 | 2.4438 |
| 19 | 7727-301-3-0175-0149 | 2552009442036732416 | 16.2885 | 4.5344 |
| 20 | 752-301-2-0324-0183 | 3689662230532099200 | 192.0995 | −0.7200 |
| 21 | 1462-301-4-0543-0079 | 3665039290398303104 | 206.7741 | 2.4244 |
| 22 | 2194-301-1-0361-0077 | 3801228644610295680 | 164.1954 | −3.5860 |
| 23 | 1462-301-4-0543-0060 | 3665226615396253824 | 206.7224 | 2.4265 |

| N | $g$ | $\sigma_g$ | $G$ | $\sigma_G$ | $\varpi_{ph}$ | $\sigma\varpi_{ph}$ | $\varpi_{tr}$ | $\sigma\varpi_{tr}$ | Sp | $M_{bol}$ |
|---|---|---|---|---|---|---|---|---|---|---|
| 1 | 20.384 | 0.021 | 19.718 | $4.3 \times 10^{-3}$ | 0.1811 | 0.0097 | 0.8123 | 0.821 | G8 | 5.3 |
| 2 | 17.674 | 0.005 | 17.260 | $8.4 \times 10^{-4}$ | 0.3086 | 0.0115 | 0.667 | 0.1211 | G0 | 4.47 |
| 3 | 17.269 | 0.005 | 16.974 | $8.7 \times 10^{-4}$ | 0.2375 | 0.0067 | 0.5096 | 0.1363 | F5 | 3.61 |
| 4 | 16.408 | 0.003 | 15.646 | $1.4 \times 10^{-3}$ | 1.4285 | 0.0592 | 1.4892 | 0.0784 | K2 | 6.08 |
| 5 | 17.335 | 0.005 | 16.882 | $7.4 \times 10^{-4}$ | 0.4255 | 0.0168 | 0.4383 | 0.1179 | G5 | 4.89 |
| 6 | 17.076 | 0.005 | 16.597 | $1.2 \times 10^{-3}$ | 0.6116 | 0.0219 | 0.6022 | 0.1014 | G8 | 5.3 |
| 7 | 17.251 | 0.005 | 16.998 | $7.8 \times 10^{-4}$ | 0.2252 | 0.0047 | 0.2149 | 0.1413 | F5 | 3.61 |
| 8 | 16.788 | 0.004 | 16.387 | $6.3 \times 10^{-4}$ | 0.4524 | 0.0191 | 0.4361 | 0.1048 | G0 | 4.47 |
| 9 | 14.645 | 0.003 | 14.512 | $6.5 \times 10^{-4}$ | 0.4975 | 0.0421 | 0.439 | 0.0439 | F2 | 2.89 |
| 10 | 16.200 | 0.003 | 15.816 | $7.9 \times 10^{-4}$ | 0.5882 | 0.0159 | 0.5176 | 0.0594 | G0 | 4.47 |
| 11 | 18.110 | 0.006 | 17.596 | $1.9 \times 10^{-3}$ | 0.3300 | 0.0066 | 0.2585 | 0.1536 | G5 | 4.89 |
| 12 | 17.327 | 0.005 | 16.824 | $1.0 \times 10^{-3}$ | 0.5571 | 0.0541 | 0.4783 | 0.1317 | G8 | 5.3 |
| 13 | 18.699 | 0.009 | 18.025 | $3.2 \times 10^{-3}$ | 0.4048 | 0.0206 | 0.3235 | 0.3079 | K0 | 5.69 |
| 14 | 17.689 | 0.005 | 17.169 | $8.9 \times 10^{-4}$ | 0.5102 | 0.0343 | 0.4149 | 0.153 | G8 | 5.3 |
| 15 | 18.616 | 0.008 | 18.106 | $2.0 \times 10^{-3}$ | 0.3095 | 0.0258 | 0.1714 | 0.217 | G8 | 5.3 |
| 16 | 18.440 | 0.007 | 17.658 | $1.2 \times 10^{-3}$ | 0.5865 | 0.0510 | 0.4417 | 0.1949 | K2 | 6.08 |
| 17 | 17.004 | 0.004 | 16.281 | $9.1 \times 10^{-4}$ | 1.0050 | 0.0601 | 0.6621 | 0.0933 | K2 | 6.08 |
| 18 | 19.437 | 0.012 | 19.017 | $4.1 \times 10^{-3}$ | 0.1455 | 0.0075 | −0.2226 | 0.4955 | G0 | 4.47 |
| 19 | 17.547 | 0.005 | 15.965 | $9.4 \times 10^{-4}$ | 3.0303 | 0.2470 | 2.6283 | 0.102 | M0 | 7.6 |
| 20 | 18.248 | 0.006 | 17.784 | $1.2 \times 10^{-3}$ | 0.2673 | 0.0174 | −0.2431 | 0.1933 | G0 | 4.47 |
| 21 | 14.820 | 0.003 | 14.332 | $4.9 \times 10^{-4}$ | 1.9047 | 0.0447 | 0.626 | 0.0398 | G8 | 5.3 |
| 22 | 19.643 | 0.015 | 17.431 | $7.7 \times 10^{-3}$ | 6.25 | 0.2734 | 4.6726 | 0.2955 | M4 | 9.92 |
| 23 | 14.901 | 0.003 | 14.361 | $3.7 \times 10^{-4}$ | 2.2988 | 0.0497 | 0.1203 | 0.0418 | K2 | 6.08 |

Stars 21 and 23 belong to red giants. Obviously, they were wrongly accepted as Main-Sequence stars in Paper18, which resulted in erroneous $\varpi_{ph}$ values.

According to Figure 4, stars 1, 2 and 3 are sub-dwarfs, and, if that is true, they have erroneous $\varpi_{ph}$ for the same reason as above. However, we pay attention to their relatively large trigonometric parallax errors (see $\sigma\varpi_{tr}$ values in Table 1), which could lead to their shift under the Main Sequence in HRD.

Lastly, stars 18 and 20 have negative parallaxes and for that reason their absolute magnitudes cannot be calculated and, consequently, they are not shown in Figure 4. Apparently the negative parallaxes indicate that these stars belong to supergiants. Again, they were wrongly accepted to be Main-Sequence stars in Paper18, which resulted in erroneous $\varpi_{ph}$ values.

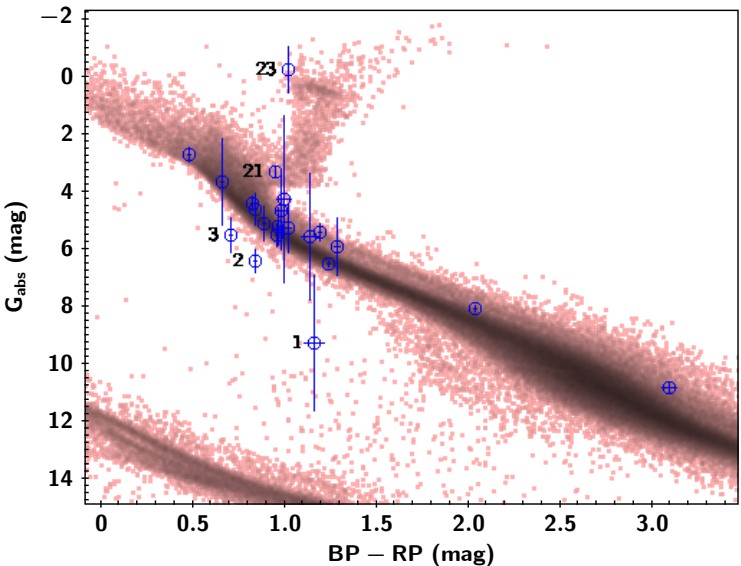

**Figure 4.** Stars used in Paper18 for the $A_V(d)$ construction (blue circles) on the HRD. Labels correspond to running numbers of stars in Table 1. Pink points are nearest ($\varpi_{tr} > 10$ mas) stars from *Gaia* DR2. Absolute magnitude $G_{abs}$ is calculated from $G_{Gaia}$ and $\varpi_{tr}$, interstellar extinction is neglected. *Gaia* G, BP and RP curves are presented in Figure 1.

The results of the comparison allow us to make the following conclusions.

1. A parameterization of stars with $\varpi_{ph} > 0.225$ mas (i.e., closer than about 4400 pc) and $g_{SDSS} < 19.^m6$ (see Figure 3) is successful, subject to a proper determination of luminosity class.
2. A rough estimate of the probability of stars belonging to the Main Sequence was carried out in Paper18 (basing on 2MASS and SDSS photometry data), and only MS stars were parameterized. Obviously, for several stars that estimation turned out to be erroneous (see Figure 4). It seems appropriate to include in the parameterization procedure information about the photometry of non-MS stars (sub-dwarfs, giants and supergiants), drawn from the literature or determined by our own efforts.

The distribution of stars used in Paper18 for the $A_V(d)$ construction by the ratio between the difference in parallax $\varpi_{ph} - \varpi_{tr}$ and the photometric parallax $\varpi_{ph}$ is presented in Figure 5. Seven outliers demonstrate $|(\varpi_{ph} - \varpi_{tr})/\varpi_{ph}| > 0.6$, they were discussed above (stars 1, 2, 3, 18, 20, 21, 23).

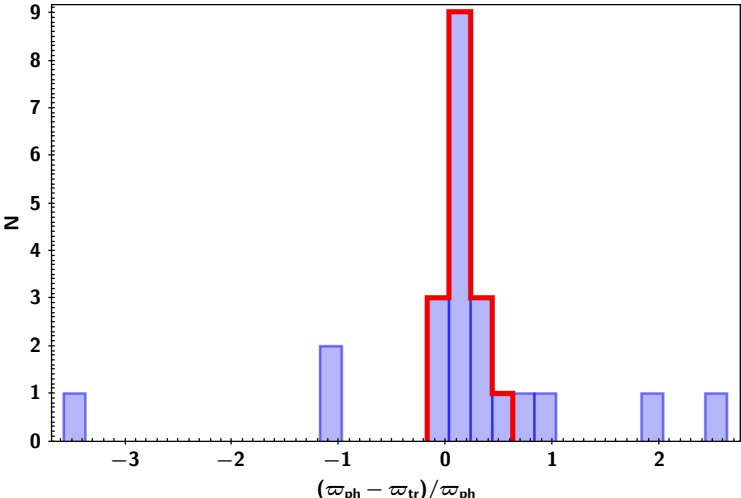

**Figure 5.** Distribution of stars used in Paper18 for the $A_V(d)$ construction by the ratio between the difference in parallax $\varpi_{ph} - \varpi_{tr}$ and the photometric parallax $\varpi_{ph}$. Subsample, designated by the red contour, does not include stars 21, 23 (red giants), 1, 2, 2 (possible sub-dwarfs) and 18, 20 (apparently supergiants), discussed in the text.

The remaining stars distribution is shown with the red contour. For them mean value of $(\varpi_{ph} - \varpi_{tr})/\varpi_{ph}$ is 0.15 with standard deviation of 0.13. Mean $\varpi_{ph}$ and $\varpi_{tr}$ values for the red contour group are 0.997 and 0.89, respectively.

It should be noted that area No 2, which showed a poor agreement with the Perlmutter et al.'s [42] data in Paper18, does not stand out in the current analysis.

It should be also noted that the cross-matching of Paper18 objects with *Gaia* DR2 catalog was made correctly. Angular distance on the sky between a Paper18 object and its *Gaia* DR2 counterpart ($\rho$) does not exceed 1 arcsec, and only for 2 of 72 objects $\rho > 0.3$ arcsec (those two stars were not selected in Paper18 for construction of $A_V(d)$ relation, and consequently, are not shown in Figures 3–5).

Based on the conclusions derived above, our procedure of parameterization of stars will be modified. In particular, the procedure will be extended to sub-dwarf, giant and supergiant stars. Also, at this stage, distant ($\varpi_{ph} < 0.225$ mas) and faint ($g_{SDSS} > 19.^m6$) objects will be removed from consideration. We will also reconstruct the procedure to use *Gaia DR2* parallaxes (and future releases, when available), as an input parameter. It is also planned to use data from other multicolor surveys, such as WISE (Wide-Field Infrared Survey Explorer), DENIS (Deep Near Infrared Survey of the Southern Sky), etc., and extend this procedure to lower galactic latitudes.

**Author Contributions:** The authors made equal contribution to this work; O.M. wrote the paper.

**Funding:** This research was partly funded by the Russian Foundation for Basic Research grant 17-52-45076. A.K. acknowledges the National Research Foundation of South Africa and the Russian Science Foundation (project no. 14-50-00043). G.Z. acknowledges support by NSFC grant No. 11390371.

**Acknowledgments:** We are grateful to our reviewers whose constructive comments greatly helped us to improve the paper. We thank E. Kilpio for fruitful collaboration. This work has made use of data from the European Space Agency (ESA) mission *Gaia* (https://www.cosmos.esa.int/gaia), processed by the *Gaia* Data Processing and Analysis Consortium (DPAC, https://www.cosmos.esa.int/web/gaia/dpac/consortium). Funding for the DPAC has been provided by national institutions, in particular the institutions participating in the *Gaia* Multilateral Agreement.

**Conflicts of Interest:** The authors declare no conflict of interest.

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
