# Peer review of "Verification of Photometric Parallaxes with Gaia DR2 Data"

_galaxies, doi:10.3390/galaxies7010007_

Reviewer 1 Report

This paper compares photometric parallaxes in some fields using several Catalogues to Gaia DR2 parallaxes. Unfortunately the paper can't be published yet as the statistical handling should be improved, correcting the parametrisation and misleading or incomplete figures.

Instead of using parallaxes, this study actually uses distances, inverting the parallaxes of objects mostly distant, some even having negative parallaxes. The paper 2018A&A...616A...9L well describes the problems which follow and also shows that truncating parallaxes or relative parallax errors introduces large biases (the 130% maximum relative error adopted here is very large). 

Working in the parallax domain would solve the issue, as eq. 2 can also be used with parallaxes rather than distances, then comparing the photometric parallaxes to the Gaia ones, without the need for any Gaia parallax truncation. 

The (asymmetrical) uncertainties which are missing on all figures would probably show that the comparisons using distances suffer from large uncertainties (and biases), another reason to use parallax instead of distance. An illustration and second problem is the conclusion "Stars with d > 3200 pc are badly parameterized (see Fig. 7)."; unfortunately, what Fig. 7 shows instead is that the ordinate (d_Gaia-d) is anticorrelated with the abscissa (d, having random errors), so the figure may rather exhibit the (expected) behaviour due to large random errors (see e.g. 2017A&A...599A..50A, fig. 47, 48). There may as well be biases due to the above-mentioned truncations, but this can't be seen in the present figures. Beside, which points are outliers can't be seen as formal uncertainties are missing. 

Correlations of random errors are probably present also in Fig. 3 and 9, but their amplitude is unknown. Even a small simulation will show what are the true systematics and what originates from random errors only. For reasons mentioned above, and also due to the good properties of the Gaia random errors, the figures should then be done using parallaxes and their uncertainties. Note that the uncertainties on the photometric estimations (2018OAst...27...62M, eq. 4?) need to be checked at small distances using the Gaia parallaxes.

The interstellar extinction which enters eq. 1 has been computed as a function of distance. As the distance comparison does not clearly prove the statistical quality of some points, the derived extinction may not be correct, and a discussion can be useful.

The abstract indicates a comparison of Gaia photometry which is actually not fully done in the paper: there is a plot, but no further analysis is done so it is unclear whether the differences are due to colour differences, binaries, x-matched errors or else. If needed, the Gaia documentation indicates relations between classical photometric bands and Gaia ones, https://gea.esac.esa.int/archive/documentation/GDR2/Data_processing/chap_cu5pho/sec_cu5pho_calibr/ssec_cu5pho_PhotTransf.html which can help the analysis.

Typos or minor points:
- l.28: "Then 26 of 251 stars with the most reliable data were selected" --> this is a very small fraction, so the the reliability criteria may be recalled.
- l. 36:  "Among 251 objects, studied in M18, only 72 were found in Gaia DR" --> with 1.7 billion objects, the very small cross-matched fraction is unlikely, except for specific types of objects. As this may imply biases due to some selection function, why the objects were not cross-matched should be indicated.
- l.47: atrometrically --> astrometrically
- l.67: what exactly means "badly parameterized" ?
- l.70: 49 pc ?
- Caption of Fig. 10: "proportional to a probability". This requires a reference, or the equation giving this probability.

Author Response

Dear reviewer,

We thank you for very useful and constructive comments, they greatly helped us to improve the paper. We have made all required corrections and responded all questions raised. All revisions are highlighted (boldfaced) in the text. The details of the revisions are given below, point-by-point.

Please, note that corrections required by other reviewers are also made and highlighted in the text.

Thank you very much again for your cooperation,
With best regards,
Oleg Malkov (on behalf of co-authors)

---------------------------------------------------------------------
Comments and Suggestions for Authors

This paper compares photometric parallaxes in some fields using several Catalogues to Gaia DR2 parallaxes. Unfortunately the paper can't be published yet as the statistical handling should be improved, correcting the parametrisation and misleading or incomplete figures. Instead of using parallaxes, this study actually uses distances, inverting the parallaxes of objects mostly distant, some even having negative parallaxes. The paper 2018A&A...616A...9L
 well describes the problems which follow and also shows that truncating parallaxes or relative parallax errors introduces large biases (the 130% maximum relative error adopted here is very large). Working in the parallax domain would solve the issue, as eq. 2 can also be used with parallaxes rather than distances, then comparing the photometric parallaxes to the Gaia ones, without the need for any Gaia parallax truncation.

===> upon your recommendation we have changed the distance domain for the parallax domain throughout the text: eq.2 is written for parallax (now it is eq.3), corresponding figures are re-drawn, numbers in conclusions are now expressed in parallax rather than distance units

The (asymmetrical) uncertainties which are missing on all figures would probably show that the comparisons using distances suffer from large uncertainties (and biases), another reason to use parallax instead of distance. An illustration and second problem is the conclusion "Stars with d > 3200 pc are badly parameterized (see Fig. 7)."; unfortunately, what Fig. 7 shows instead is that the ordinate (d_Gaia-d) is anticorrelated with the abscissa (d, having random errors), so the figure may rather exhibit the (expected) behaviour due to large random errors (see e.g. 2017A&A...599A..50A, fig. 47, 48).
 There may as well be biases due to the above-mentioned truncations, but this can't be seen in the present figures. Beside, which points are outliers can't be seen as formal uncertainties are missing.

===> Fig.7 is re-plotted (now it is Fig.3, left panel) and, consequently, the conclusions are re-written.

Correlations of random errors are probably present also in Fig. 3 and 9, but their amplitude is unknown. Even a small simulation will show what are the true systematics and what originates from random errors only. For reasons mentioned above, and also due to the good properties of the Gaia random errors, the figures should then be done using parallaxes and their uncertainties. Note that the uncertainties on the photometric estimations (2018OAst...27...62M, eq. 4?) need to be checked at small distances using the Gaia parallaxes.

===> Figs. 3 and 9 are removed, the remaining figures contain errors

The interstellar extinction which enters eq. 1 has been computed as a function of distance. As the distance comparison does not clearly prove the statistical quality of some points, the derived extinction may not be correct, and a discussion can be useful.

===> distances, computed in M18 (changed now for Paper18, acting on an advice of another reviewer), are now transferred into parallaxes to correctly compare values with Gaia DR2 data

The abstract indicates a comparison of Gaia photometry which is actually not fully done in the paper: there is a plot, but no further analysis is done so it is unclear whether the differences are due to colour differences, binaries, x-matched errors or else. If needed, the Gaia documentation indicates relations
 between classical photometric bands and Gaia ones, https://gea.esac.esa.int/archive/documentation/GDR2/Data_processing/chap_cu5pho/sec_cu5pho_calibr/ssec_cu5pho_PhotTransf.html
 which can help the analysis.

===> you are right, comparison of photometry is not done, and we actually use Gaia photometry only for illustration of correctness of the Paper18-DR2 cross-matching. The abstract is corrected.

Typos or minor points:
- l.28: "Then 26 of 251 stars with the most reliable data were selected" --> this is a very small fraction, so the the reliability criteria may be recalled.

===> a detailed explanation is now provided (after Eq.3)

- l. 36:  "Among 251 objects, studied in M18, only 72 were found in Gaia DR" --> with 1.7 billion objects, the very small cross-matched fraction is unlikely,
 except for specific types of objects. As this may imply biases due to some selection function, why the objects were not cross-matched should be indicated.

===> a vast majority of our objects are too faint to be registered in Gaia DR2 (only 20% of Paper18 objects are brighter than gSDSS=19.5). It is now
inserted in the text.

- l.47: atrometrically --> astrometrically

===> corrected

- l.67: what exactly means "badly parameterized" ?

===> that means that one can not rely upon results of parameterization. This and two next sentences are re-phrased.

- l.70: 49 pc ?

===> it should be corrected: "49 pc" -> "49", but that fragment is now removed from the text

- Caption of Fig. 10: "proportional to a probability". This requires a reference, or the equation giving this probability.

===> Fig.10 is removed from the manuscript
---------------------------------------------------------------------

Reviewer 2 Report

General comments:

The main problem of this paper is the fact that is not considering Bailer-Jones+2018 paper to transform parallaxes into distances, considering the parallax uncertainties. I think this is a big issue here as the paper is suposed to concentrate on parallaxes and they should be treated correctly. All the analysis should be repeated considering a better procedure to determine distances from Gaia parallaxes and do not remove negative parallaxes from the sample. The error bars should also be analysed and plotted in the figures to see how comparable the different distance estimates are and what is the reason for the discrepancy among them. This analysis should be done in the paper and stated in the conclusions. No clues are provided to justify the reason of the discrepancies. It would be nice if based on the results in this paper some preliminar modification in the M18 method is attempted and show if an improvement could be indicated based on that.

On the other hand, the writing is often too vague, imprecise and hard to follow. Methods should be better explained in the paper (in a self-consistent way and not simply citing external papers). Results and conclusions should be quantified numerically and not qualitatively, for instance as "badly" or "successfully".

The paper is too short and with very few details provided. It looks like if a maximum number of pages were imposed to the authors (maybe some proceedings?). If this is the case, the referee is not aware of this and maybe some comments about the low level of description could be understood, although the text would benefit with more details anyway. Related with this, the paper seems to strongly rely on M18 paper and the knowledge of the user of it. Some of the variables used are even not described in the text (sigma, P, ...) and let the reader to interpret themself and it is hard to follow.

The caption of the Figures should be written in a different way, not simply "A vs B". It makes very difficult to interpret them at first sight and creates confusion to the reader. Units of the represented magnitudes should appear in all axes labels (between parenthesis). Some Figures could be joined in a single one to make the reader to access more easily the different plots for comparison without looking at different pages. In the text, the conclusions derived from every Figure should be stated more clearly to justify its presence in the paper and the information that they provide. Error bars in the plots should be included.

Along the paper, first person in the text is used (WE used, provide US, ...). I would use impersonal form (were used, provided ...)

Abstract: I would expand the abstract to be more descriptive of the work inside the paper (context, methods and results) and avoid vague expressions like "under certain circumstances" and explain which are those circumstances.

1. Introduction

Line 9: "good dispersion and sufficient accuracy". Do you mean "good signal-to-noise ratio"?

Line 10: The sentence about the need of large telescopes to observe faint objects is quite obvious. I think the writing of this paragraph should be improved better explaining the differences between photometry and spectroscopy.

Line 11: I would remove the sentence "It was preliminary studied in [1]" and just include the reference to [1] in the previous sentence.

Line 13: "provide us" --> "provide"

Lines 14-15: Last sentence of the paragraph is not obvious. An indication of how this can be done should be provided.

Line 17: "allows us to use its data for solving" --> "allows to be used for solving"

Line 18: "verify" --> "improve"?

Line 21: "involvment" --> "involvement"

2. Data and method

This section should be expanded. A summary of the methodology in M18 reference should be provided as it is frequently cited in the paper and it should be as self-consistent as possible.

By the way, the use of M18 as abreviation is not very convenient, as it can be mislead with Messier 18 open cluster. It could simply be cited as [3] all along the paper?

- First paragraph: "in 2MASS" --> "with 2MASS"

- Crossmatch with 2MASS and SDSS is provided in GDR2. The authors do not need to repeat the crossmatch themselves. Maybe problems with the epoch used (Gaia is not using J2000.0, but J2015.5) in this crossmatching could appear if not done properly.

- Define meaning of "MK" (Morgan-Keenan).

- Why only 251 objects are considered (Gaia has more than one billion sources with parallaxes). How these sources were selected? The list of these sources should be provided in order to allow the reader to reproduce your results.

- "SpT", "d" and "Av" should be separated with commas or parenthesis in the text. 

- When mentioning 13 photometric bands, detail which are these 13 bands. (In Fig. 1 only 8 passbands are shown. Later on it is said that only G and g are considered.).

- Equation 2 explicits that M_i A_i are a fucntion of the SpT and Av, respectively. I think this should not be included in the equation but later on, in the text, when defining each term. For instance "A_i=f(Av) is the interstellar extincion law, and Mi=f(SpT) is the absolute magnitude...". Anyway I don't see the point to explicit this dependency even in the text, as these magnitudes also depend on some other factors, not only those indicated. For example, Mi depends also on the luminosity class, or Ai also on the colour excess. 

Line 25: The observational errors of the magnitudes are usually called with a \sigma symbol, and not a \Delta.

Line 26: From Av which law was used to retrieve A_i in any other passband? Cardelli+1989? Fitzpatrick+2011? Others? And what is dependency you used from SpT to derive Mi?

Line 28: What does it mean "the most reliable data". Provide numbers and details.

Line 31: Reference (or details and numerical criteria if done by the authors) is needed after each "agreement" in the line. All these "agreement" analysis is done using the methods explained by the authors in the paper or independently? How these comparisons were done?

Line 32: Reference to supernovae to confirm the accelerated expansion of the Universe sounds out of topic and should be better explained. Parenthesis about No 2 area is not understood. If a table or plot with the different coordinates of the sky regions is included, then we will know what No 2 means.

Line 34: "the question, posed in Introduction". Repeat which is the question you refer to.

Line 35: What are the coordinates in the sky of the four sky regions analysed in the paper?

Line 41: As said, Bailer-Jones+2018 method (or analogous considering sigma_pi) should not be ignored by the authors.

Line 43: The use of the "simplest estimate" is not suitable in Gaia era and something more sophisticated should be used.

Line 45: Have the Gaia DR2 known issues also been considered? (see https://www.cosmos.esa.int/web/gaia/dr2-known-issues)

Figure 1: Text in section 2 mentions 13 photometric bands. This figure only shows 8. And later on in the text only 2 are used (G and g). What's the point of showing these passbands here? You should provide the reference to the passbands in the caption of the figure.

Line 50: Why only g and G were finally chosen? In fact G is more similar to r than to g passband.

Line 52: Figure 3 says that sigmapi/pi<2.8 and here 1.3 is said. Use only one value to not mislead the reader (although I understand that there are no sources in your sample between 1.3 and 2.8).

Figure 2: Caption should be rewritten to better understand the content of the plot and the meaning of the different lines/symbols. Units in the axes labels should be provided. Add "as a black line" after "is indicated" in the first sentence. I would firstly explain that all points are those stars in M18 found in DR2 and then explain the different colours as satisfying of not ACS. Appendix A in Evans+2018 details the colour relationship between G and g. Then the colour of the sources (and not only its magnitude) are relevant to understand its behaviour in the plot. Maybe you could substitute this plot by some colour-colour diagram (G-g vs BP-RP or G-g vs g-i).

Figure 3: Labels should be improved (plx means nothing) and units should be provided. "2.8" value in the caption should agree with value in the text (see Line 52 comment). Negative parallaxes are not removed from the plot (although it could be interpreted according with the text in the caption). They are removed in the analysis (but they shouldn't).

Paragraph starting with "Concluding this Section..." should be moved to the end of the document or when crossmatching is mentioned (although I would prefer to first show the plots with the main results before discussing minor details like this to justify the main results). Here it is out of context. Anyway, the use of Figs 4 and 5 to justify that the cross-matching procedure was correct is not completely clear to me.

Line 56: "Concluding this section we should note" --> "It should be noted"

Line 58: "demonstrates correlation" --> "demonstrate no correlation"

Line 60: Nomenclature for distances is misleading from here (even more when using also angular distance). I would use \Delta d instead of D (also in line 65). The name d_{Gaia} seems to indicate that these distances were provided by Gaia team in the release and this is not the case. These distances were derived by the authors (and not according to Gaia team guidelines). I would, then, change the name of this variable. On the other hand "d" should be named as "d_M18" or something similar, because it makes the impression that "d" is the real distance. More details about how each distance is derived should be provided in the text to understand and interpret the differences among them.

Figure 5: The meaning of horizontal axes should be clearly indicated. At first sight it seems that it shows negative distances (which would not have physical meaning), but they are negative differences instead. To avoid confusion I would show Fig. 6 before Fig. 5.

Lines 67, 68 and 79: What is the meaning of "badly parameterized". Provide numbers to judge how "badly parameterized" they are. What is the possible reason for this bad result?

Line 69: Define the meaning of \sigma. The reference to 1 is not clear. It is a reference to paper [1]? Or Fig. 1?. \sigma=\sigma_pi? From GDR2?

Item 4: 

Line 71: What are the stars "that satisfy the criteria 1 and 2"? the bad cases or the good ones?

Figures 6-10: "are given" --> "as". Why there are some blue circles decentered from the green points? Rewrite the captions explaining the meaning of D and d more clearly. d should be named as dM18. Define P plotted in Fig. 10. For a better reading these different figures could be joined in one or two plots with right and left panels.

Line 84: "when available". Gaia parallaxes are currently available already.

Line 85: What information would add the other multicolor surveys? What they will allow to improve?

References:

- References should be ordered in alphabetical order.

- Very long list of authors is not nice. The use of "et al" formula should be used.

Author Response

Dear reviewer,

We thank you for very useful and constructive comments, they greatly helped us to improve the paper. We have made all required corrections and responded all questions raised. All revisions are highlighted (boldfaced) in the text. The details of the revisions are given below, point-by-point.

Please, note that corrections required by other reviewers are also made and highlighted in the text.

Thank you very much again for your cooperation,
With best regards,
Oleg Malkov (on behalf of co-authors)

---------------------------------------------------------------------
Comments and Suggestions for Authors

General comments:

The main problem of this paper is the fact that is not considering Bailer-Jones+2018 paper to transform parallaxes into distances, considering the parallax uncertainties. I think this is a big issue here as the paper is suposed to concentrate on parallaxes and they should be treated correctly. All the analysis should be repeated considering a better procedure to determine distances from Gaia parallaxes and do not remove negative parallaxes from the sample.
 The error bars should also be analysed and plotted in the figures to see how comparable the different distance estimates are and what is the reason for the discrepancy among them. This analysis should be done in the paper and stated in the conclusions. No clues are provided to justify the reason of the discrepancies.

===> acting on the advice of you and another reviewer, we have decided to waive the distances from our analysis and base our conclusions on the parallaxes. The majority of figures are re-drawn and corresponding conclusions are re-written. The error bars are added in the figures.

It would be nice if based on the results in this paper some preliminar modification in the M18 method is attempted and show if an improvement could be indicated based on that.

===> we describe possible modifications in the M18 method (see the last paragraph of the manuscript), but we plan to modify the method after publication of the current paper

On the other hand, the writing is often too vague, imprecise and hard to follow. Methods should be better explained in the paper (in a self-consistent way and not simply citing external papers).

===> the "Introduction" and "Data and method" sections are significantly expanded

Results and conclusions should be quantified numerically and not qualitatively, for instance as "badly" or "successfully".

===> results and conclusions are re-phrased, in particular the terms "badly" and "successfully" are removed or explained

The paper is too short and with very few details provided. It looks like if a maximum number of pages were imposed to the authors (maybe some proceedings?). If this is the case, the referee is not aware of this and maybe some comments about the low level of description could be understood, although the text would benefit with more details anyway. Related with this, the paper seems to strongly rely on M18 paper and the knowledge of the user of it. Some of the variables used are even not described in the text (sigma, P, ...) and let the reader to interpret themself and it is hard to follow.

===> indeed, the paper is based on a presentation at a conference, however, no page limit was imposed for us. Now the paper is significantly expanded. In particular, important fragments of M18 is inserted in the current paper. The variable sigma (now substituted by "D") is explained in Eq.1, and the variable P (and corresponding figure) is removed from the text.

The caption of the Figures should be written in a different way, not simply "A vs B". It makes very difficult to interpret them at first sight and creates confusion to the reader. Units of the represented magnitudes should appear in all axes labels (between parenthesis). Some Figures could be joined in a single one to make the reader to access more easily the different plots for comparison without looking at different pages. In the text, the conclusions derived from every Figure should be stated more clearly to justify its presence in the paper and the information that they provide. Error bars in the plots should be included.

===> the captions are rewritten, units of the represented magnitudes appear in axes labels between parenthesis, and error bars are included in the plots.
Two figs are joined (now it is Fig.3). The conclusions are corrected.

Along the paper, first person in the text is used (WE used, provide US, ...). I would use impersonal form (were used, provided ...)

===> about 25 corrections are made in the text

Abstract: I would expand the abstract to be more descriptive of the work inside the paper (context, methods and results) and avoid vague expressions
 like "under certain circumstances" and explain which are those circumstances.

===> Abstract is re-written, specific values for limiting distance and brightness are given in the text

1. Introduction

Line 9: "good dispersion and sufficient accuracy". Do you mean "good signal-to-noise ratio"?

===> it is now explained in Introduction

Line 10: The sentence about the need of large telescopes to observe faint objects is quite obvious. I think the writing of this paragraph should be improved
 better explaining the differences between photometry and spectroscopy.

===> see above

Line 11: I would remove the sentence "It was preliminary studied in [1]" and just include the reference to [1] in the previous sentence.

===> done (now it is the reference to [17])

Line 13: "provide us" --> "provide"

===> corrected

Lines 14-15: Last sentence of the paragraph is not obvious. An indication of how this can be done should be provided.

===> the sentence is kept, but Introduction is re-written, so now it should be obvious enough

Line 17: "allows us to use its data for solving" --> "allows to be used for solving"

===> corrected

Line 18: "verify" --> "improve"?

===> corrected

Line 21: "involvment" --> "involvement"

===> re-phrased

2. Data and method

This section should be expanded. A summary of the methodology in M18 reference should be provided as it is frequently cited in the paper and it should be as self-consistent as possible.

===> the "Data and method" sections is significantly expanded

By the way, the use of M18 as abreviation is not very convenient, as it can be mislead with Messier 18 open cluster. It could simply be cited as [3] all along the paper?

===> M18 is changed for Paper18, it is advisable to separate this reference from others, as it is cited repeatedly

- First paragraph: "in 2MASS" --> "with 2MASS"

===> corrected

- Crossmatch with 2MASS and SDSS is provided in GDR2. The authors do not need to repeat the crossmatch themselves. Maybe problems with the epoch used (Gaia is not using J2000.0, but J2015.5) in this crossmatching could appear if not done properly.

===> we have checked, our crossmatch is correct. Besides 2MASS and SDSS, we should cross-match Gaia DR2 objects with GALEX, UKIDSS and probably other surveys in future.

- Define meaning of "MK" (Morgan-Keenan).

===> corrected

- Why only 251 objects are considered (Gaia has more than one billion sources with parallaxes). How these sources were selected? The list of these sources should be provided in order to allow the reader to reproduce your results.

===> 251 objects were selected to study in Paper18 before the Gaia DR2 appears, and a vast majority of these objects are too faint to be registered in Gaia DR2 (only 20% of Paper18 objects are brighter than gSDSS=19.5). This explanation is now inserted in the text.

- "SpT", "d" and "Av" should be separated with commas or parenthesis in the text.

===> done

- When mentioning 13 photometric bands, detail which are these 13 bands. (In Fig. 1 only 8 passbands are shown. Later on it is said that only G and g are considered.).

===> (actually 14) photometric bands are now listed in the text.

- Equation 2 explicits that M_i A_i are a fucntion of the SpT and Av, respectively. I think this should not be included in the equation but later on, in the text, when defining each term. For instance "A_i=f(Av) is the interstellar extincion law, and Mi=f(SpT) is the absolute magnitude...". Anyway I don't see the point to explicit this dependency even in the text, as these magnitudes also depend on some other factors, not only those indicated. For example, Mi depends also on the luminosity class, or Ai also on the colour excess.

===> the functions are excluded from Eq.2 (and Eq.3), and included in the text. MK SpT includes the luminosity class.

Line 25: The observational errors of the magnitudes are usually called with a \sigma symbol, and not a \Delta.

===> corrected

Line 26: From Av which law was used to retrieve A_i in any other passband? Cardelli+1989? Fitzpatrick+2011? Others? And what is dependency you used from SpT to derive Mi?

===> two corresponding paragraphs with citations are inserted in the text

Line 28: What does it mean "the most reliable data". Provide numbers and details.

===> this is now explained and re-phrased in the text

Line 31: Reference (or details and numerical criteria if done by the authors) is needed after each "agreement" in the line. All these "agreement" analysis
 is done using the methods explained by the authors in the paper or independently? How these comparisons were done?

===> references to corresponding sections of Paper18 is now inserted in the text

Line 32: Reference to supernovae to confirm the accelerated expansion of the Universe sounds out of topic and should be better explained. Parenthesis about No 2 area is not understood. If a table or plot with the different coordinates of the sky regions is included, then we will know what No 2 means.

===> all data are presented in Table 2 of Paper18, we do not reproduce it here, but give a reference to the table.

Line 34: "the question, posed in Introduction". Repeat which is the question you refer to.

===> it is re-phrased now

Line 35: What are the coordinates in the sky of the four sky regions analysed in the paper?

===> coordinates are now given in the beginning of Section "Data and method"

Line 41: As said, Bailer-Jones+2018 method (or analogous considering sigma_pi) should not be ignored by the authors.

===> following to your and another reviewer's recommendation, we removed distances from our analysis and deal with parallaxes

Line 43: The use of the "simplest estimate" is not suitable in Gaia era and something more sophisticated should be used.

===> we do not use distances more, so our "simplest estimate" is not actual

Line 45: Have the Gaia DR2 known issues also been considered? (see https://www.cosmos.esa.int/web/gaia/dr2-known-issues)

===> we have read this with interest. Principles for the object selection (the ACS filter) are improved, but it has little impact on our conclusions, as they are made for stars used in Paper18 for the $A_V(d)$ construction regardless whether or not they pass the ACS filter.

Figure 1: Text in section 2 mentions 13 photometric bands. This figure only shows 8. And later on in the text only 2 are used (G and g). What's the point of showing these passbands here? You should provide the reference to the passbands in the caption of the figure.

===> in our study Paper18, we used photometry from 2MASS, SDSS, GALEX and UKIDSS (altogether 14 bands, they are listed now after Eq.1). Five of them (ugriz from SDSS) are shown in Fig.1 together with Gaia bands (the Gaia bands were not used in Paper18). We have inserted references in the caption of Fig.1

Line 50: Why only g and G were finally chosen? In fact G is more similar to r than to g passband.

===> we had not made attempts to achieve a similarity in passbands, and g(SDSS) passband was chosen because the majority of our objects are brighter in g than in u,r,i,z. [We have removed photometry comparison from the text at all now.]

Line 52: Figure 3 says that sigmapi/pi<2.8 and here 1.3 is said. Use only one value to not mislead the reader (although I understand that there are no sources in your sample between 1.3 and 2.8).

===> it is now removed from the text

Figure 2: Caption should be rewritten to better understand the content of the plot and the meaning of the different lines/symbols. Units in the axes labels should be provided.

===> units in the axes labels are provided

 Add "as a black line"  after "is indicated" in the first sentence.

===> corrected

 I would firstly explain that all points are those stars in M18 found in DR2 and then explain the different colours as satisfying of not ACS.

===> corrected

 Appendix A in Evans+2018 details the colour relationship between G and g. Then the colour of the sources (and not only its magnitude) are relevant to understand its behaviour in the plot. Maybe you could substitute this plot by some colour-colour diagram (G-g vs BP-RP or G-g vs g-i).

===> we have substituted that mag-mag diagram by HRD (see Fig.4), and indeed this helped us to understand some stars' behaviour

Figure 3: Labels should be improved (plx means nothing) and units should be provided. "2.8" value in the caption should agree with value in the text
 (see Line 52 comment). Negative parallaxes are not removed from the plot (although it could be interpreted according with the text in the caption).
 They are removed in the analysis (but they shouldn't).

===> Fig.3 disappeared, but labels on other figures are improved, and units are provided. Negative parallaxes are kept in the further analysis.

Paragraph starting with "Concluding this Section..." should be moved to the end of the document or when crossmatching is mentioned (although I would prefer to first show the plots with the main results before discussing minor details like this to justify the main results). Here it is out of context. Anyway, the use of Figs 4 and 5 to justify that the cross-matching procedure was correct is not completely clear to me.

===> that paragraph is moved to the end of the manuscript, and Figs. 4 and 5 are removed

Line 56: "Concluding this section we should note" --> "It should be noted"

===> corrected

Line 58: "demonstrates correlation" --> "demonstrate no correlation"

===> that phrase is removed

Line 60: Nomenclature for distances is misleading from here (even more when using also angular distance). I would use \Delta d instead of D (also in line 65). The name d_{Gaia} seems to indicate that these distances were provided by Gaia team in the release and this is not the case. These distances were derived by the authors (and not according to Gaia team guidelines). I would, then, change the name of this variable. On the other hand "d" should be named as "d_M18" or something similar, because it makes the impression that "d" is the real distance. More details about how each distance is derived should be provided in the text to understand and interpret the differences among them.

===> we do not deal with distances now (however, I agree with all your considerations)

Figure 5: The meaning of horizontal axes should be clearly indicated. At first sight it seems that it shows negative distances (which would not have physical meaning), but they are negative differences instead. To avoid confusion I would show Fig. 6 before Fig. 5.

===> Fig.5 is removed, and Fig.6 (distance comparison) is substituted by parallax comparison (currently Fig.2)

Lines 67, 68 and 79: What is the meaning of "badly parameterized". Provide numbers to judge how "badly parameterized" they are. What is the possible reason for this bad result?

===> the Section is re-written, and all "badly parameterized" are removed

Line 69: Define the meaning of \sigma. The reference to 1 is not clear. It is a reference to paper [1]? Or Fig. 1?. \sigma=\sigma_pi? From GDR2?

===> actually it was \sigma from Eq.1, but now symbols for parameters are changed. Also, that particular item is removed from the "conclusions" list.

Item 4:

Line 71: What are the stars "that satisfy the criteria 1 and 2"? the bad cases or the good ones?

===> it is removed

Figures 6-10: "are given" --> "as". Why there are some blue circles decentered from the green points? Rewrite the captions explaining the meaning of D and d more clearly. d should be named as dM18. Define P plotted in Fig. 10. For a better reading these different figures could be joined in one or two plots with right and left panels.

===> "are given" are not used more. Indeed some blue circles decentered from the green points, now it is corrected. D and d are not used more (in the sense 'distance').

Line 84: "when available". Gaia parallaxes are currently available already.

===> it is now changed for "Gaia DR2 parallaxes (and future releases, when available)"

Line 85: What information would add the other multicolor surveys? What they will allow to improve?

===> according to our expectations, the more the number of photometric bands, the better observational photometry reproduces SED, and, consequently,
the higher of the quality of determined parameters (SpT, distance, extinction, ...)

References:

- References should be ordered in alphabetical order.

- Very long list of authors is not nice. The use of "et al" formula should be used.

===> Presentation of the References section (in particular, the order of appearance of references) is determined by the Bibliography style for MDPI journals, which deals with bibtex entries, provided by authors. We have taken the bibtex entries for our bibliography from ADS. So the number of the authors in the list is determined both by ADS and MDPI.
---------------------------------------------------------------------

Reviewer 3 Report

This paper entitled "Verification of photometric parallaxes with Gaia DR2 data" aims to propose a new parametrisation for the star distances and their interstellar extinctions using a very small subset of stars in common within the Malkov+18 sample and Gaia DR2. Although the overall idea of this work is very nice and will have a great impact in this flourishing field, I found the paper very qualitative in both the description and robustness of the results. My major concern regards the sample selection that sometimes results very confusing. In addition, a solid statistical analysis of the results is missing as well as a proper treatment of the errors related to the different results. Moreover, the readability of the paper is sometimes compromised by poor grammar and interrupted flow of ideas. The author should therefore consider getting a native English speaker to proof read the paper.

In the report below, I have noted several points in which further clarification and/or discussion is required. I believe the authors should address these points before I can consider the manuscript for publication in the Galaxies journal.

ABSTRACT

1) I was wondering if there is any restriction regarding the length of the abstract because I found it too vague. If not please 1) spell out all the acronyms in there, 2) list the "certain circumstances" in detail 3) specify what is "high-latitude".

INTRODUCTION

2) General comments on this section: I found this section a bit incomplete. The authors should explain what is the state-of-the-art of this topic; add more references to this section and conclude the section with the outline of the paper. In particular, my feeling is that each sentence is lacking of proper credits to previous works. I am not providing any preferential references but I would let the author to rewrite this section in a more comprehensive way.

Page 1
Line 8 "Parameters of a star" --> be more specific and add some reference here
Line 8 "spectrum" --> Which kind of spectrum the author are thinking about? Optical, infrared, UV, several wavelength ranges?
Line 10 "large telescope" --> How much large? Please be more precise...
Line 10 "bright enough" --> Please provide a lower limit for the luminosity or AB magnitudes.
Line 12 "large photometric surveys" --> Please mention at least a couple of these surveys
Line 18 "a number of stellar astronomy problems" --> Again, please mention which are these problems.
Line 20 "In this paper" --> In this paper,
Line 21 "involvment of" --> including the

DATA AND METHOD
3) Regarding the interstellar extinction computed for the four selected areas. How robust is this assumption? Have the authors tried to compare the individual interstellar extinction values with the one used for the full area and not just with the LAMOST and supernovae values? In the case of LAMOST and supernovae comparison, please quantify how good is the agreement, i.e. providing the percentage or Chi_2 values.

4) What do the authors mean with the sentence "(the issue with the remaining area, No 2, is still open)"?

5) The authors claim they want "To answer the question, posed in Introduction" but there is no clear question in the mentioned section

6) Sample selection: this is one of my major comment... It is not clear how many stars will be finally used to obtain the parametrisation. I strongly suggest to describe better the sample selection even including a summary table.

7) Fig.1: Please use the same thickness for the Gaia filters lines. Add the parenthesis for the x axis units and include (A. U.) for the y-axis ones.

8) The authors are considering the SDSS photometric data. How do they deal with the well known aperture effects?

9) Fig.2: Please add the error bars and as I said before, the sample is very confusing. It is not clear what are the criteria for the sample selection, as far as I understood, there some stars in common to M18, others that satisfy the ACS requirements and then the blue circles are the ones used for the parametrisation, is this right? I encourage also to add a legend to the plot.

10) On the basis of which arguments do you choose the 1.3 value to exclude points?

Page 2
Line 24 four selected areas in the sky --> For completeness the authors should summarise which are these fields by giving the coordinates
Line 24 "2MASS, SDSS, GALEX, and UKIDSS" --> Spell out these acronyms
Line 24 "combined" --> panchromatic or multi wavelength
Line 28 "most reliable" --> What do the authors mean with reliable? With more photometric data available or with lowest uncertainties related to the data? Please clarify this statement.
Line 30 "LAMOST" --> spell out
Line 47 "atrometrically" --> astrometrically
Line 48 "distances" --> distances,
Lina 50 "work" --> work,

Page 4

11) Fig.3: Again the error bars are missing, in the caption is written that the error bars are plotted but unless they are smaller than the points, it is impossible to see them... Once more also here the sample is very confusing, it is almost impossible to identify the points that are plotted in Fig. 2. Therefore, I suggest to keep the same colour/symbol for the stars that have been considered for the parametrisation...

12) The authors claim "the cross-matching of M18 objects with Gaia DR2 catalogue was made correctly.", on the basis of which kind of statistical test? I would suggest to perform some KS or AD tests or any other statistical test they consider appropriated.

Line 56 "this Section"---> this Section,

13) Fig.4 and 5: Please drawn the error bars and complete the caption explicitly explaining what the axis are.  On the other hand, my suggestion is to merge this two figures in one.

Page 5
RESULTS AND CONCLUSIONS

14) I encourage the authors to rewrite this section adding some discussion to the results. In particular, I feel that Fig.s from 6 to 10 are poorly commented in the main text. For each one of the results I will clearly state **why** is the case of such claim and not just given a shopping list.

Line 65 "different parameters" --> Please mention here the parameters...

15)I strongly suggest to revise the style of Fig.s 6 to 10 on the basis of my comments to Fig.2

16) Due to the fact that this section gives also the main conclusions of the paper, the author should clearly state again here what are D, d, g and sigma

Lines 82 and 84 For clarity, please spell out the MS, WISE and DENIS acronyms

Page 8
Line 76 Perlmutter et al’s —> Perlmutter et al.’s
Line 81 In particular —> In particular,

Pages 9 and 10
REFERENCE
Why for some papers the authors provide the arXiv number and for others not?

Author Response

Dear reviewer,

We thank you for very useful and constructive comments, they greatly helped us to improve the paper. We have made all required corrections and responded all questions raised. All revisions are highlighted (boldfaced) in the text. The details of the revisions are given below, point-by-point.

Please, note that corrections required by other reviewers are also made and highlighted in the text.

Thank you very much again for your cooperation,
With best regards,
Oleg Malkov (on behalf of co-authors)

---------------------------------------------------------------------
Comments and Suggestions for Authors

This paper entitled "Verification of photometric parallaxes with Gaia DR2 data" aims to propose a new parametrisation for the star distances and their interstellar extinctions using a very small subset of stars in common within the Malkov+18 sample and Gaia DR2. Although the overall idea of this work is very nice and will have a great impact in this flourishing field, I found the paper very qualitative in both the description and robustness of the results. My major concern regards the sample selection that sometimes results very confusing. In addition, a solid statistical analysis of the results is missing as well as a proper treatment of the errors related to the different results. Moreover, the readability of the paper is sometimes compromised by poor grammar and interrupted flow of ideas. The author should therefore consider getting a native English speaker to proof read the paper.

===> one of co-authors, a native English speaker, carefully checked the paper

In the report below, I have noted several points in which further clarification and/or discussion is required. I believe the authors should address these points before I can consider the manuscript for publication in the Galaxies journal.

ABSTRACT

1) I was wondering if there is any restriction regarding the length of the abstract because I found it too vague. If not please 1) spell out all the acronyms in there,

===> the acronyms are spelled out in Abstract and repeated in the beginning of the section "Data and method"

 2) list the "certain circumstances" in detail

===> specific values for limiting distance and brightness are now given

 3) specify what is "high-latitude".

===> limiting galactic latitude value is now given

INTRODUCTION

2) General comments on this section: I found this section a bit incomplete. The authors should explain what is the state-of-the-art of this topic; add more references to this section and conclude the section with the outline of the paper. In particular, my feeling is that each sentence is lacking of proper credits to previous works. I am not providing any preferential references but I would let the author to rewrite this section in a more comprehensive way.

===> the section is significantly expanded, but I boldface in the text only specific corrections, see below

Page 1
Line 8 "Parameters of a star" --> be more specific and add some reference here

===> corrected

Line 8 "spectrum" --> Which kind of spectrum the author are thinking about? Optical, infrared, UV, several wavelength ranges?

===> corrected

Line 10 "large telescope" --> How much large? Please be more precise...

===> specification is added in the text

Line 10 "bright enough" --> Please provide a lower limit for the luminosity or AB magnitudes.

===> estimations are given in the text

Line 12 "large photometric surveys" --> Please mention at least a couple of these surveys

===> corrected

Line 18 "a number of stellar astronomy problems" --> Again, please mention which are these problems.

===> corrected

Line 20 "In this paper" --> In this paper,

===> corrected

Line 21 "involvment of" --> including the

===> corrected

DATA AND METHOD
3) Regarding the interstellar extinction computed for the four selected areas. How robust is this assumption? Have the authors tried to compare the individual interstellar extinction values with the one used for the full area and not just with the LAMOST and supernovae values? In the case of LAMOST and supernovae comparison, please quantify how good is the agreement, i.e. providing the percentage or Chi_2 values.

===> the paragraph is rephrased

4) What do the authors mean with the sentence "(the issue with the remaining area, No 2, is still open)"?

===> the phrase is re-written to make it clearer

5) The authors claim they want "To answer the question, posed in Introduction" but there is no clear question in the mentioned section

===> the beginning of the phrase is re-written

6) Sample selection: this is one of my major comment... It is not clear how many stars will be finally used to obtain the parametrisation. I strongly suggest to describe better the sample selection even including a summary table.

===> a detailed explanation, containing the selection principles description, is inserted in the beginning of Section 2, after Eqs (1), (2) and (3). A table is now included.

7) Fig.1: Please use the same thickness for the Gaia filters lines. Add the parenthesis for the x axis units and include (A. U.) for the y-axis ones.

===> done

8) The authors are considering the SDSS photometric data. How do they deal with the well known aperture effects?

===> we believe that the aperture effect (i) does not affect a majority of the selected (not very bright and not very faint) stars, and (ii) results in corresponding flags in the SDSS catalogue (see my answer on your item 6 above)

9) Fig.2: Please add the error bars and as I said before, the sample is very confusing. It is not clear what are the criteria for the sample selection, as far as I understood, there some stars in common to M18, others that satisfy the ACS requirements and then the blue circles are the ones used for the parametrisation, is this right? I encourage also to add a legend to the plot.

===> former Fig.2 (Photometry comparison) is now removed, among some other figures, and the error bars are added to the remaining and new ones.
You are right about the meaning of the symbols, the caption is re-written. However, I find it difficult to prepare a compact enough legend for the plot. M18 is changed now for Paper18, acting on an advice of another reviewer.

10) On the basis of which arguments do you choose the 1.3 value to exclude points?

===> that fragment is now excluded from the text

Page 2
Line 24 four selected areas in the sky --> For completeness the authors should summarise which are these fields by giving the coordinates

===> the coordinates are added in the beginning of Section "Data and method"

Line 24 "2MASS, SDSS, GALEX, and UKIDSS" --> Spell out these acronyms

===> acronyms are spelled out ibid

Line 24 "combined" --> panchromatic or multi wavelength

===> changed to multi wavelength

Line 28 "most reliable" --> What do the authors mean with reliable? With more photometric data available or with lowest uncertainties related to the data? Please clarify this statement.

===> it is now explained, and "most reliable" is removed from the text

Line 30 "LAMOST" --> spell out

===> spelled out

Line 47 "atrometrically" --> astrometrically

===> corrected

Line 48 "distances" --> distances,

===> corrected

Lina 50 "work" --> work,

===> corrected

Page 4

11) Fig.3: Again the error bars are missing, in the caption is written that the error bars are plotted but unless they are smaller than the points, it is impossible to see them... Once more also here the sample is very confusing, it is almost impossible to identify the points that are plotted in Fig. 2. Therefore, I suggest to keep the same colour/symbol for the stars that have been considered for the parametrisation...

===> Fig.3 is removed from the text. For identification of (at least some) points we have added labels, corresponding the running numbers in the Table.

12) The authors claim "the cross-matching of M18 objects with Gaia DR2 catalogue was made correctly.", on the basis of which kind of statistical test?
 I would suggest to perform some KS or AD tests or any other statistical test they consider appropriated.

===> We do not perform any statistical test, we base our conclusion on the fact that all but two our (Paper18) stars are found in DR2 with positional accuracy (rho) better than 0.3 arcsec, and rho correlates neither with photometric (gSDSS-gGaia) difference, nor with distance (distancePaper18-distanceGaia) difference. The two exceptions were not selected in Paper18 for the A_V(d) construction.

Line 56 "this Section"---> this Section,

===> corrected

13) Fig.4 and 5: Please drawn the error bars and complete the caption explicitly explaining what the axis are.  On the other hand, my suggestion is to merge this two figures in one.

===> Fig.4 and 5 are removed

Page 5
RESULTS AND CONCLUSIONS

14) I encourage the authors to rewrite this section adding some discussion to the results. In particular, I feel that Fig.s from 6 to 10 are poorly commented in the main text. For each one of the results I will clearly state **why** is the case of such claim and not just given a shopping list.

===> the Section is substantially re-written

Line 65 "different parameters" --> Please mention here the parameters...

===> parameters are mentioned now: photometric parallax and SDSS-g magnitude

15) I strongly suggest to revise the style of Fig.s 6 to 10 on the basis of my comments to Fig.2

===> revised

16) Due to the fact that this section gives also the main conclusions of the paper, the author should clearly state again here what are D, d, g and sigma

===> these particular parameters do not more appear in the text, but meaning of all other parameters is explained here

Lines 82 and 84 For clarity, please spell out the MS, WISE and DENIS acronyms

===> WISE and DENIS acronyms are spelled out here, MS is spelled out earlier in the text (here I substituted "non-MS" for "sub-dwarf, giant and supergiant" that is more precisely in this context)

Page 8
Line 76 Perlmutter et al's -> Perlmutter et al.'s

===> corrected

Line 81 In particular -> In particular,

===> corrected

Pages 9 and 10
REFERENCE
Why for some papers the authors provide the arXiv number and for others not?

===> Presentation of the References section is determined by the Bibliography style for MDPI journals, which deals with bibtex entries, provided by authors. We have taken the bibtex entries for our bibliography from ADS. For some papers ADS provides the arXiv number, and for others not. I assume that in the latter case the authors just did not upload their manuscripts to arXiv.
---------------------------------------------------------------------

Round  2

Reviewer 1 Report

The authors have followed the recommandations in their new manuscript. Now that the uncertainties are present, there looks to be an underestimation of the photometric uncertainties in Table 1, as the relative error on the parallax is 4% on the average, and can be as low as 2%, which is hard to believe (in the magnitude domain this would probably be much to small compared to the main sequence width plus the extinction uncertainty). A more conservative estimate should be given, and checked thanks to comparisons to Gaia at small distances.

- l. 12: gien star --> given star
- l. 81: Obvioualy --> Obviously
- l. 95: multucolor --> multicolor
- l. 116: varpi
- l.v151:  on this stage --> at this stage

Author Response

Dear reviewer,

Thank you very much for the careful reading of the revised version of the paper and for your new set of comments. All suggested corrections are made, but, with your permission, I did not boldface the corrections in the text, as they are mostly typos and grammatical errors. The details of the revisions are given below.

Again, please, note that corrections required by other reviewers are also made in the text.

On behalf of my co-authors I thank you once more for your help. Your comments greatly helped us to improve the paper.

Sincerely yours,
Oleg Malkov
---------------------------------------------------------------------
Comments and Suggestions for Authors

The authors have followed the recommandations in their new manuscript. Now that the uncertainties are present, there looks to be an underestimation
 of the photometric uncertainties in Table 1, as the relative error on the parallax is 4% on the average, and can be as low as 2%, which is hard to believe
 (in the magnitude domain this would probably be much to small compared to the main sequence width plus the extinction uncertainty). A more conservative estimate should be given, and checked thanks to comparisons to Gaia at small distances.

===> Indeed, the relative error on the photometric parallax is underestimated. In the current study observational photometric errors are considered to be
the only source for the resulting parameters errors. Consequently, here we underestimate the error values. To calculate errors more correctly, one should take into account also calibration tables errors and relations errors. It is discussed in Paper18, and we have inserted this explanation in the text (after the first paragraph of the Section "Results and conclusions").

- l. 12: gien star --> given star
- l. 81: Obvioualy --> Obviously
- l. 95: multucolor --> multicolor
- l. 116: varpi
- l. 151:  on this stage --> at this stage

===> everything is corrected
---------------------------------------------------------------------

Reviewer 2 Report

Congratulation to the authors for their effort incorporating the previous comments. The paper is in better shape now. I have enjoyed now the discussion using the HR diagram helping to understand the behaviour of the outliers and providing clues for future refinements in the methodology.

I understand it as a comparsion of the behaviour of old data with respect to the new observations. But I recommend that in the future you are not stuck to the old set of sources and try to expand as much as possible your sample, starting from Gaia whole dataset and select suitable sources. This will easily provide thousand of sources instead of just the very few you are considering now. Also the work in Paper18 should be redone under these premises, as you are fitting laws using only 5 datapoints!! Also remember to use Bailer-Jones+2018 methodology to distance estimations.

Why sources need to be so faint? Cannot this study be done with brighter sources?

In several places in the text (abstract, for instance, among other places) you are still using the word "distance", although you say in your answer to previous comments that parallaxes were analysed instead, and not distances.

About the bibliography. Is it possible to include the year of the publication in the text, and not only the authors? I guess this is driven by the style template from the magazine. For instance, without the year is not understood why Paper18 is called like this.

Gaia should be in italic letters all along the document.

Abstract:

Remove "estimation of distance"

1. Introduction:

- "parameters of a gien star" --> "parameters of a given star"

- IR and UV should be defined.

- Period dot is needed before "Spectral observations..."

- "R=10000" --> "R=100\, 000"

- "...interstellar extinction is the estimation of extinction (as well..." This sentence have two times the word "extinction" and it is not needed. It could be "...interstellar extinction is its estimation (as well..."

- References to SDSS and GALEX should be added when mentioned in line 30.

- "Consequently, it allows user" --> "Consequently, it allows the user"

- Reference to Kilpio&Malkov[19] is better after "was made"

- "and it was found that the investigated maps ofter demonstrate contradictory results." --> "and contradictory results were found."

- "like determination of distance to stars". distances are not determined now?

- "the verification of a method" --> "the verification of the method and stellar sample analysed in Paper18"

2. Data and method

- "The (l,b) coordinates of the areas are (334,+61.9)" --> "The galactic coordinates of the areas are (l,b)=(334,+61.9)". A plot of the distribution of sources in the sky could be nice.

- To avoid confusion between distance "d" and "D" I would call "D" as "\chi", because it is the usual chi-squared. On the other hand I would avoid "d" and use "parallax" instead, as you are now not using distances, right?

- " over up to 14 photometric bands" --> "over up to N=14 photometric bands"

- Equation 3 is not correct when using observational data. The mean value of the parallax is not enough and their errors should be considered to derive a good value for the distance to be substituted in Eq. 2 to derive Eq. 3 (see Bailer-Jones+2018).

- "Here mobs,j and sigmamobs,i are the apparent magnitude..." This sentence should be right after Eq.1 and not after Eq. 3.

- Ai is not the "interstellar extinction law" but the extinction in the ith photometric band

- I would split the sentence defining Ai and Mi. After mentioning Ai explain how to retrieve Ai from Av. Then define Mi and explain how it can be obtained through the tables.

- Remove "the following studies: data for" and simply say "...used data from 2MASS..."

- Provide reference to 2MASS, SDSS and UKIDSS

- "GALEX were taken from Yuan et al [34]" --> "GALEX (Yuan et al [34])

- "Both teams have made calculations for SDSS photometry" --> "Both teams made Ai calculations for SDSS photometry"

- "... our comparison show a very good agreement between their results" --> Show a plot or something justifying this sentence.

- "Altogether 251 objects were found" ---> I still thing this is too few number of objects. The potential number of sources is Gaia are thousands-million sources. Say what is the initial number of sources before the crossmatch. You should try to increase the number of sources to populate your sample and derive more meaningful conclusions.

- "the original surveys contain various flags" --> "the original surveys contain various flags which allow us to remove unsuitable objects".

- "Av is assumed to be within 0.5 mag" --> "Av is assumed to be smaller than 0.5 mag"

- "d is assumed to be within 8000 pc" --> "d is assumed to be closer than 8000 pc".

- Remove "Obviously". It is not obvious that negative values should be removed. Uncertainties should be considered. Explain also here why negative values could appear. (Section 5 in Paper18 does not explain it. It only explain possible sources of discrepancies, not the presence of negative values)

- "see Fig. 5 in Paper18" --> Fig 5 in Paper18 shows only a cloud of 5 points, without their error bars and a line plotted on the top of it, but the linear behaviour is not very clear with only these 5 points. Populating this plot with more points (from Gaia) would be very good for this study and not relying only on this 5 points to derive a relationship.

- Extrapolation of the poor Av(d) relationship to infinity is not recommended. Justify why this is possible and the uncertainty that this could introduce.

- "from multucolor photometry" --> "from multicolor photometry"

- You are still removing negative parallaxes and should not be removed.

- You are always using Av(d), but should be Av(parallax)?

- Still not clear why Fig 1 only plots SDSS and Gaia passbands and not all 14 passbands. And also why only g and G are finally used ignoring all other 12 passbands.

Font size in axis labels in Fig. 3 should be increased to be clearer for the reader.

- Define variables in line 116 of the text (relative parallax uncertainty, varpitr, ...)

Table 1. No need to use subscript in GSDSS and GGaia passbands. Simply use g and G as column header detailing in the caption that g correspond to SDSS and G to Gaia.

Author Response

Dear reviewer,

Thank you very much for the careful reading of the revised version of the paper and for your new set of comments. All suggested corrections are made, but, with your permission, I did not boldface the corrections in the text, as they are mostly typos and grammatical errors. The details of the revisions are given below, point-by-point.

Again, please, note that corrections required by other reviewers are also made in the text.

On behalf of my co-authors I thank you once more for your help. Your comments greatly helped us to improve the paper.

Sincerely yours,
Oleg Malkov

---------------------------------------------------------------------
Comments and Suggestions for Authors

Congratulation to the authors for their effort incorporating the previous comments. The paper is in better shape now. I have enjoyed now the discussion using the HR diagram helping to understand the behaviour of the outliers and providing clues for future refinements in the methodology.

I understand it as a comparsion of the behaviour of old data with respect to the new observations. But I recommend that in the future you are not stuck to the old set of sources and try to expand as much as possible your sample, starting from Gaia whole dataset and select suitable sources. This will easily provide thousand of sources instead of just the very few you are considering now. Also the work in Paper18 should be redone under these premises,
 as you are fitting laws using only 5 datapoints!! Also remember to use Bailer-Jones+2018 methodology to distance estimations.

===> Thank you for the recommendations. That is exactly what we planned to do.

Why sources need to be so faint? Cannot this study be done with brighter sources?

===> Our main intention was to construct a procedure for determination of interstellar extinction - distance (Av(d)) relation for small areas on the sky.
Bright sources are rare, and we deal with, say, 0.1 degree areas, where we are sure that Av(d) is identical for the whole area.

In several places in the text (abstract, for instance, among other places) you are still using the word "distance", although you say in your answer to previous comments that parallaxes were analysed instead, and not distances.

===> "distance" is corrected for "parallax" in some places, but in other cases usage of "distance" is justified.

About the bibliography. Is it possible to include the year of the publication in the text, and not only the authors? I guess this is driven by the style template from the magazine. For instance, without the year is not understood why Paper18 is called like this.

===> Yes, I agree, but indeed it is driven by the style template from the magazine.

Gaia should be in italic letters all along the document.

===> Corrected.

===> All comments below are taken into account, and corresponding corrections are made in the text.

Abstract:

Remove "estimation of distance"

1. Introduction:

- "parameters of a gien star" --> "parameters of a given star"

- IR and UV should be defined.

- Period dot is needed before "Spectral observations..."

- "R=10000" --> "R=100\, 000"

- "...interstellar extinction is the estimation of extinction (as well..." This sentence have two times the word "extinction" and it is not needed. It could be "...interstellar extinction is its estimation (as well..."

- References to SDSS and GALEX should be added when mentioned in line 30.

- "Consequently, it allows user" --> "Consequently, it allows the user"

- Reference to Kilpio&Malkov[19] is better after "was made"

- "and it was found that the investigated maps ofter demonstrate contradictory results." --> "and contradictory results were found."

- "like determination of distance to stars". distances are not determined now?

- "the verification of a method" --> "the verification of the method and stellar sample analysed in Paper18"

2. Data and method

- "The (l,b) coordinates of the areas are (334,+61.9)" --> "The galactic coordinates of the areas are (l,b)=(334,+61.9)". A plot of the distribution of sources in the sky could be nice.

- To avoid confusion between distance "d" and "D" I would call "D" as "\chi", because it is the usual chi-squared. On the other hand I would avoid "d" and use "parallax" instead, as you are now not using distances, right?

- " over up to 14 photometric bands" --> "over up to N=14 photometric bands"

- Equation 3 is not correct when using observational data. The mean value of the parallax is not enough and their errors should be considered to derive a good value for the distance to be substituted in Eq. 2 to derive Eq. 3 (see Bailer-Jones+2018).

- "Here mobs,j and sigmamobs,i are the apparent magnitude..." This sentence should be right after Eq.1 and not after Eq. 3.

- Ai is not the "interstellar extinction law" but the extinction in the ith photometric band

- I would split the sentence defining Ai and Mi. After mentioning Ai explain how to retrieve Ai from Av. Then define Mi and explain how it can be obtained through the tables.

- Remove "the following studies: data for" and simply say "...used data from 2MASS..."

- Provide reference to 2MASS, SDSS and UKIDSS

- "GALEX were taken from Yuan et al [34]" --> "GALEX (Yuan et al [34])

- "Both teams have made calculations for SDSS photometry" --> "Both teams made Ai calculations for SDSS photometry"

- "... our comparison show a very good agreement between their results" --> Show a plot or something justifying this sentence.

- "Altogether 251 objects were found" ---> I still thing this is too few number of objects. The potential number of sources is Gaia are thousands-million sources. Say what is the initial number of sources before the crossmatch. You should try to increase the number of sources to populate your sample and derive more meaningful conclusions.

- "the original surveys contain various flags" --> "the original surveys contain various flags which allow us to remove unsuitable objects".

- "Av is assumed to be within 0.5 mag" --> "Av is assumed to be smaller than 0.5 mag"

- "d is assumed to be within 8000 pc" --> "d is assumed to be closer than 8000 pc".

- Remove "Obviously". It is not obvious that negative values should be removed. Uncertainties should be considered. Explain also here why negative values could appear. (Section 5 in Paper18 does not explain it. It only explain possible sources of discrepancies, not the presence of negative values)

- "see Fig. 5 in Paper18" --> Fig 5 in Paper18 shows only a cloud of 5 points, without their error bars and a line plotted on the top of it, but the linear behaviour is not very clear with only these 5 points. Populating this plot with more points (from Gaia) would be very good for this study and not relying only on this 5 points to derive a relationship.

- Extrapolation of the poor Av(d) relationship to infinity is not recommended. Justify why this is possible and the uncertainty that this could introduce.

- "from multucolor photometry" --> "from multicolor photometry"

- You are still removing negative parallaxes and should not be removed.

- You are always using Av(d), but should be Av(parallax)?

- Still not clear why Fig 1 only plots SDSS and Gaia passbands and not all 14 passbands. And also why only g and G are finally used ignoring all other 12 passbands.

Font size in axis labels in Fig. 3 should be increased to be clearer for the reader.

- Define variables in line 116 of the text (relative parallax uncertainty, varpitr, ...)

Table 1. No need to use subscript in GSDSS and GGaia passbands.
 Simply use g and G as column header detailing in the caption
 that g correspond to SDSS and G to Gaia.
---------------------------------------------------------------------

Reviewer 3 Report

I thank the Authors for their detailed answers and for their further work. The authors addressed most of my comments in the previous report. I still have some minor suggestions that should be taken into account before I can recommend this paper for publication. 

GENERAL
I am sorry for the native English co-author but I still find some typos and grammatical errors, please proof-read the manuscript more carefully.

INTRODUCTION

Line 10: is in --> is
Line 10: there are too many "of the", I would suggest to change "of the surface" with "belonging to the surface"
Line 12: gien --> given
Line 15: accurancy —> accuracy
Line 15: Add a "." after sufficient accuracy
Line 15: Spectral observations --> For instance, spectroscopic observations
Line 31: their objects? --> Maybe the authors means several objects
Lines 34-35: Kilpio and Malkov [19] was made --> was made by Kilpio and Malkov [19]. It was...
Lines 36-37: can be compiled [20], Hakkila et al. [21], Malkov and Kilpio [22]. --> Please rephrase this sentence, is not clear what the authors want to say here
Line 51: of a method of parameterization of stars --> too many "of"
Line 53 for parameterization for which --> too many "for"

DATA AND METHOD
Line 57: (l,b) --> Please explicitly say which kind of coordinates and in which system/units they are given
Line 67: of stars of --> for stars of
Line 80: demonstrated --> presented
Line 81: Obvioualy --> Obviously
Lines 81-82: with possible reasons for such solutions discussed --> This sentence is a bit confusing... can you please rephrase it?
Line 95: multucolor --> multicolor

RESULTS AND CONCLUSIONS
Line 124: At last --> Lastly,
Line 145: It should be noted also --> It should be also noted
Lines 149-150: of parameterization of stars and determination of interstellar extinction --> too many "of"
Lines 155-156: the procedure application --> this procedure

Author Response

Dear reviewer,

Thank you very much for the careful reading of the revised version of the paper and for your new set of comments. All suggested corrections are made, but, with your permission, I did not boldface the corrections in the text, as they are mostly typos and grammatical errors.

Again, please, note that corrections required by other reviewers are also made in the text.

On behalf of my co-authors I thank you once more for your help. Your comments greatly helped us to improve the paper.

Sincerely yours,
Oleg Malkov